# Glutamine deprivation triggers NAGK-dependent hexosamine salvage

Sydney Campbell[1,2], Clementina Mesaros[3], Luke Izzo[1,2], Hayley Affronti[1,2], Michael Noji[1,2], Bethany E Schaffer[4], Tiffany Tsang[1,2], Kathryn Sun[5], Sophie Trefely[1,2,6], Salisa Kruijning[1,2], John Blenis[4], Ian A Blair[3], Kathryn E Wellen[1,2]*

[1]Department of Cancer Biology, University of Pennsylvania, Philadelphia, United States; [2]Abramson Family Cancer Research Institute, University of Pennsylvania, Philadelphia, United States; [3]Department of Systems Pharmacology and Translational Therapeutics, University of Pennsylvania, Philadelphia, United States; [4]Meyer Cancer Center and Department of Pharmacology, Weill Cornell Medicine, New York, United States; [5]Pancreatic Cancer Research Center, Perelman School of Medicine, University of Pennsylvania, Philadelphia, United States; [6]Center for Metabolic Disease Research, Department of Microbiology and Immunology, Lewis Katz School of Medicine, Temple University, Philadelphia, United States

**Abstract** Tumors frequently exhibit aberrant glycosylation, which can impact cancer progression and therapeutic responses. The hexosamine biosynthesis pathway (HBP) produces uridine diphosphate N-acetylglucosamine (UDP-GlcNAc), a major substrate for glycosylation in the cell. Prior studies have identified the HBP as a promising therapeutic target in pancreatic ductal adenocarcinoma (PDA). The HBP requires both glucose and glutamine for its initiation. The PDA tumor microenvironment is nutrient poor, however, prompting us to investigate how nutrient limitation impacts hexosamine synthesis. Here, we identify that glutamine limitation in PDA cells suppresses de novo hexosamine synthesis but results in increased free GlcNAc abundance. GlcNAc salvage via N-acetylglucosamine kinase (NAGK) is engaged to feed UDP-GlcNAc pools. *NAGK* expression is elevated in human PDA, and *NAGK* deletion from PDA cells impairs tumor growth in mice. Together, these data identify an important role for NAGK-dependent hexosamine salvage in supporting PDA tumor growth.

*For correspondence:
wellenk@upenn.edu

## Introduction

Altered glycosylation is frequently observed in malignancies, impacting tumor growth as well as immune and therapeutic responses (*Akella et al., 2019*; *Mereiter et al., 2019*; *Munkley, 2019*). Several types of glycosylation, including O-GlcNAcylation and N-linked glycosylation, are dependent on the glycosyl donor uridine diphosphate N-acetylglucosamine (UDP-GlcNAc), which is synthesized by the hexosamine biosynthesis pathway (HBP). The HBP branches off from glycolysis with the transfer of glutamine's amido group to fructose-6-phosphate (F-6-P) to generate glucosamine-6-phosphate (GlcN-6-P), mediated by the rate limiting enzyme glutamine—fructose-6-phosphate transaminase (GFPT1/2). The pathway further requires acetyl-CoA, ATP, and uridine triphosphate (UTP) to ultimately generate UDP-GlcNAc. O-GlcNAcylation, the addition of a single N-acetylglucosamine (GlcNAc) moiety onto a serine or threonine residue of intracellular proteins, is upregulated in multiple cancers (*Akella et al., 2019*). Targeting O-GlcNAcylation suppresses the growth of breast, prostate, and colon cancer tumors (*Caldwell et al., 2010*; *Ferrer et al., 2017*; *Gu et al., 2010*; *Guo et al., 2017*; *Lynch et al., 2012*). Similarly, highly branched N-glycan structures are sensitive to

**eLife digest** Inside tumors, cancer cells often have to compete with each other for food and other resources they need to survive. This is a key factor driving the growth and progression of cancer. One of the resources cells need is a molecule called UDP-GlcNAc, which they use to modify many proteins so they can work properly. Because cancer cells grow quickly, they likely need much more UDP-GlcNAc than healthy cells.

Many tumors, including those derived from pancreatic cancers, have very poor blood supplies, so their cells cannot get the nutrients and other resources they need to grow from the bloodstream. This means that tumor cells have to find new ways to use what they already have. One example of this is developing alternative ways to obtain UDP-GlcNAc. Cells require a nutrient called glutamine to produce UDP-GlcNAc. Limiting the supply of glutamine to cells allows researchers to study how cells are producing UDP-GlcNAc in the lab.

Campbell et al. used this approach to study how pancreatic cancer cells obtain UDP-GlcNAc when their access to glutamine is limited. They used a technique called isotope tracing, which allows researchers to track how a specific chemical is processed inside the cell, and what it turns into. The results showed that the pancreatic cancer cells do not make new UDP-GlcNAc but use a protein called NAGK to salvage GlcNAc (another precursor of UDP-GlcNAc), which may be obtained from cellular proteins. Cancer cells that lacked NAGK formed smaller tumors, suggesting that the cells grow more slowly because they cannot recycle UDP-GlcNAc fast enough.

Pancreatic cancer is one of the most common causes of cancer deaths and is notable for being difficult to detect and treat. Campbell et al. have identified one of the changes that allows pancreatic cancers to survive and grow quickly. Next steps will include examining the role of NAGK in healthy cells and testing whether it could be targeted for cancer treatment.

HBP flux and are upregulated in malignant tissue (*Lau et al., 2007*), and targeting the relevant Golgi GlcNAc transferase enzymes can limit tumor growth and metastasis in vivo (*Granovsky et al., 2000*; *Li et al., 2008*; *Zhou et al., 2011*). Thus, improved understanding of the regulation of the HBP in cancer could point towards novel therapeutic strategies.

Pancreatic ductal adenocarcinoma (PDA) is a deadly disease with a 5 year survival rate of 9% and a rising number of annual deaths (*Rahib et al., 2014*) (ACS Cancer Facts and Figures 2019, NIH SEER report 2019). Mutations in *KRAS* occur in nearly all cases of human PDA and drive extensive metabolic reprogramming in cancer cells. Enhanced flux into the HBP was identified as a primary metabolic feature mediated by mutant KRAS in PDA cells (*Ying et al., 2012*). Hypoxia, a salient characteristic of the tumor microenvironment (*Lyssiotis and Kimmelman, 2017*), was shown to further promote expression of glycolysis and HBP genes in pancreatic cancer cells (*Guillaumond et al., 2013*). Notably, the glutamine analog 6-diazo-5-oxo-L-norleucine (DON), which inhibited the HBP, suppressed PDA metastasis, and sensitized PDA tumors to anti-PD1 therapy (*Sharma et al., 2020*). DON has also been reported to sensitize PDA cells to the chemotherapeutic gemcitabine in vitro (*Chen et al., 2017*). Additionally, a recently developed inhibitor targeting the HBP enzyme phosphoacetylglucosamine mutase 3 (PGM3) enhances gemcitabine-mediated reduction of xenograft tumor growth in vivo (*Ricciardiello et al., 2020*). Thus, the HBP may represent a therapeutic target in PDA, although the regulation of UDP-GlcNAc synthesis and the optimal strategies to target this pathway for therapeutic benefit in PDA remain poorly understood.

An outstanding question is the impact of the tumor microenvironment on UDP-GlcNAc synthesis. The HBP has been proposed as a nutrient-sensing pathway since its rate-limiting step, mediated by GFPT1/2, requires both glutamine and the glycolytic intermediate fructose-6-phosphate (*Denzel and Antebi, 2015*). In hematopoietic cells, glucose deprivation limits UDP-GlcNAc levels and dramatically reduces levels of the N-glycoprotein IL3Rα at the plasma membrane in a manner dependent on the HBP (*Wellen et al., 2010*). Similarly, O-GlcNAcylation of certain nuclear-cytosolic proteins, including cancer-relevant proteins such as Myc and Snail, has been demonstrated to be nutrient sensitive, impacting protein stability or function (*Housley et al., 2008*; *Park et al., 2010*; *Swamy et al., 2016*). Yet, the PDA tumor microenvironment is thought to be particularly nutrient poor, owing to its characteristic dense stroma (*Halbrook and Lyssiotis, 2017*). This raises the

question of how nutrient deprivation impacts the synthesis of UDP-GlcNAc and its utilization for glycosylation. Understanding how PDA cells regulate these processes under nutrient limitation could identify therapeutic vulnerabilities. In this study, we investigated the impact of nutrient deprivation on the HBP and glycosylation in PDA cells, identifying a key role for hexosamine salvage through the enzyme N-acetylglucosamine kinase (NAGK) in PDA tumor growth.

## Results

### Tetra-antennary N-glycans and O-GlcNAcylation are minimally impacted by nutrient limitation in pancreatic cancer cells

To examine the effects of nutrient deprivation on glycosylation, we cultured cells under glucose or glutamine limitation and examined O-GlcNAc levels and cell surface phytohemagglutinin-L (L-PHA) binding, a readout of N-acetylglucosaminyltransferase 5 (MGAT5)-mediated cell surface N-glycans (*Figure 1—figure supplement 1A,B*), which are highly sensitive to UDP-GlcNAc availability (*Lau et al., 2007*). We focused on glucose and glutamine because of their requirement to initiate the HBP (*Figure 1A*). First, as a positive control, we examined HCT-116 and SW480 colon cancer cells, previously documented to have glucose-responsive O-GlcNAcylation (*Park et al., 2010*; *Steenackers et al., 2016*), which we also confirmed in HCT-116 cells (*Figure 1B*). Indeed, L-PHA binding was suppressed by glucose restriction in SW480 cells and by glutamine restriction in both colon cancer cell lines (*Figure 1C*). Next, to test whether glycans were sensitive to nutrient restriction in PDA cells, we examined L-PHA binding and O-GlcNAc levels under nutrient deprivation conditions in a panel of human PDA cell lines, including PANC-1, MIA PaCa-2, AsPC-1, and HPAC. Across these cell lines, no consistent changes in L-PHA binding were observed under glucose or glutamine limitation (*Figure 1D,E*, *Figure 1—figure supplement 1C*). We also examined L-PHA binding in PDA cells under oxygen- or serum-deprived conditions and observed minimal changes (*Figure 1—figure supplement 1D,E*). O-GlcNAcylation was minimally altered by culture in low glutamine and exhibited variable changes in response to glucose limitation (*Figure 1F*), consistent with stress-induced regulation of this modification (*Taylor et al., 2008*). Since glycosylation may be maintained through either sustained ability to add the modifications or through changes in turnover, we assayed active O-GlcNAcylation by inhibiting O-GlcNAcase with Thiamet G (TMG). TMG treatment resulted in equivalently elevated O-GlcNAcylation levels in high and low glutamine conditions (*Figure 1G*; *Figure 1—figure supplement 1F*), indicating that glutamine restriction does not limit the capacity of cells to add the O-GlcNAc modification. Mia-PaCa-2 cells exhibited some cell death in low glutamine, though this was not exacerbated by TMG treatment (*Figure 1—figure supplement 1F*). Thus, under a variety of nutrient stress conditions, neither L-PHA binding nor O-GlcNAcylation were consistently suppressed in pancreatic cancer cell lines. Glutamine restriction in particular had remarkably little impact on O-GlcNAcylation and L-PHA binding, raising the question of how UDP-GlcNAc is generated during nutrient limitation.

### De novo UDP-GlcNAc synthesis is suppressed upon glutamine limitation

We therefore next asked whether the abundance of HBP metabolites is impacted by nutrient limitation. We measured HBP metabolites after glucose or glutamine restriction using HPLC-MS (*Guo et al., 2016b*). In low glutamine conditions, GlcN-6-P levels were potently decreased relative to glutamine-replete conditions in PANC-1 cells, while UDP-GlcNAc abundance was maintained (*Figure 2A*). In MIA PaCa-2 cells, UDP-GlcNAc abundance actually increased upon glutamine restriction (*Figure 2—figure supplement 1A*). These data indicate that UDP-GlcNAc might be generated through mechanisms other than de novo synthesis. Glycolytic intermediates were minimally impacted by low glutamine conditions, and TCA cycle intermediates such as α-KG and malate decreased as expected (*Figure 2A*, *Figure 2—figure supplement 1A*). In contrast to that in glutamine restriction, UDP-GlcNAc abundance declined in 5 mM or 0.1 mM relative to 10 mM glucose conditions (*Figure 2—figure supplement 1B*), suggesting that glutamine limitation specifically may trigger an adaptive response to sustain UDP-GlcNAc pools.

We sought to understand how UDP-GlcNAc pools are sustained during glutamine restriction. In addition to de novo synthesis of UDP-GlcNAc via the HBP, free GlcNAc in the cell can also be phosphorylated via N-acetylglucosamine kinase (NAGK) to produce GlcNAc-P and then regenerate UDP-

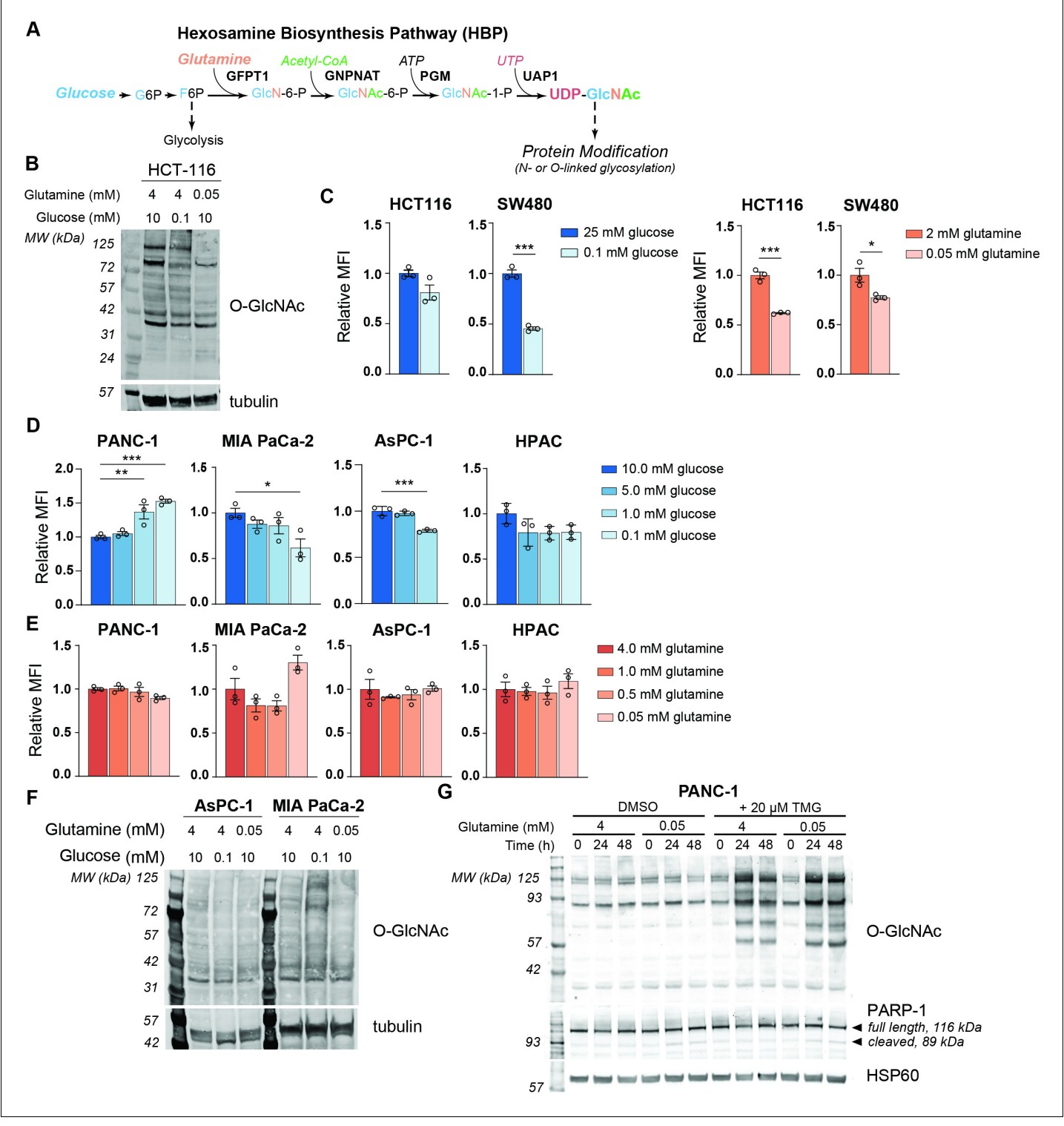

**Figure 1.** MGAT5-dependent N-glycans are minimally impacted by glucose or glutamine deprivation in PDA cells. (**A**) Overview of the hexosamine biosynthesis pathway (HBP). (**B**) O-GlcNAc levels in HCT-116 cells in high and low nutrients; cells were incubated in indicated concentrations of glucose and glutamine for 48 hr. (**C**) Phytohemagglutinin-L (L-PHA) binding in colon cancer cells. Cells were incubated in the indicated concentrations of glucose (left) or glutamine (right) for 48 hr and then analyzed by flow cytometry. Graph shows mean fluorescence intensity (MFI) relative to control condition. Statistical significance was calculated by unpaired t-test. (**D, E**) L-PHA binding in pancreatic ductal adenocarcinoma (PDA) cells in low nutrients. Cells were incubated in the indicated concentrations of glucose (**D**) or glutamine (**E**) for 48 hr and then analyzed by flow cytometry. Statistical significance was calculated by one-way ANOVA. (**F**) O-GlcNAc levels in PDA cells in high and low nutrients. Cells were incubated in the indicated

*Figure 1 continued on next page*

*Figure 1 continued*

concentrations of glutamine for 48 hr. (G) Western blot for O-GlcNAc in PANC-1 cells cultured in the indicated concentrations of glutamine with or without Thiamet-G (TMG) treatment for the indicated time. For all bar graphs, mean ± standard error of the mean (SEM) of three biological replicates is represented. Panels (B–G) are representative of at least two independent experimental replicates. *p≤0.05; **p≤0.01; ***p≤0.001.

The online version of this article includes the following figure supplement(s) for figure 1:

**Figure supplement 1.** L-PHA binding detects MGAT5-dependent glycans.

GlcNAc (*Figure 2B*). However, NAGK's roles in physiology and cancer biology have been minimally studied. To investigate the possibility that UDP-GlcNAc is generated through mechanisms other than its synthesis from glucose, we first designed a stable isotope labeling strategy to quantify the fraction of the glucosamine ring that is synthesized de novo in glutamine-replete versus -restricted conditions. Since multiple components of UDP-GlcNAc [glucosamine ring, acetyl group, uridine (both the uracil nucleobase and the ribose ring)] can be synthesized from glucose, UDP-GlcNAc iso-topologs up to M+16 can be generated from glucose (*Moseley et al., 2011*; *Figure 2C*). In order to measure the glucose carbon incorporated into GlcNAc-P and UDP-GlcNAc via the HBP, all isotopo-logs containing a fully labeled glucosamine ring are added together (% labeled GlcN indicates sum of M+6, M+8, M+11, and M+13 for UDP-GlcNAc and sum of M+6 and M+8 for GlcNAc-P) (*Figure 2C*).

After 48 hr of glutamine restriction, cells were incubated with fresh low glutamine medium con-taining [U-$^{13}$C]-glucose to track the incorporation of glucose carbons into hexosamine intermediates. Across multiple PDA cell lines, the fractional labeling of the glucosamine ring in both GlcNAc-P and UDP-GlcNAc pools was markedly suppressed by glutamine restriction, indicating decreased de novo synthesis in low glutamine conditions (*Figure 2D*, *Figure 2—figure supplement 1C–E*). Notably, labeling into the ribose component of UDP-GlcNAc was also suppressed (% labeled ribose indicates sum of isotopologs containing M+5 [i.e., M+5, M+7, M+11, and M+13]; *Figure 2D*, *Figure 2—fig-ure supplement 1C*). Consistently, incorporation of $^{13}$C glucose into UTP was suppressed upon glu-tamine restriction (*Figure 2—figure supplement 2A*), even though UTP levels were maintained or increased (*Figure 2A*, *Figure 2—figure supplement 1A*), suggesting a role for nucleoside salvage in maintaining nucleotide pool in these conditions. This is consistent with previous reports demonstrat-ing that autophagy/ribophagy is a source of nucleosides in amino acid-deprived conditions (*Guo et al., 2016a*; *Wyant et al., 2018*). Indeed, silencing of either of the uridine salvage enzymes uridine kinase 1 or 2 (UCK1/2) resulted in decreased UDP, UTP, and UDP-GlcNAc levels (*Figure 2—figure supplement 2B,C*), indicating that nucleoside salvage contributes to maintaining uridine phosphate and UDP-GlcNAc pools. Thus, glutamine restriction suppresses the de novo synthesis of both GlcNAc-P and UTP, both of which are required to produce UDP-GlcNAc.

We noted that GlcNAc-P and UDP-GlcNAc pools labeled from glucose with similar but not identi-cal kinetics. While this is potentially due to limitations in detection since GlcNAc-P is much less abun-dant than UDP-GlcNAc, we considered whether GlcNAc-P-independent pathways may also have minor contributions to glucose-dependent UDP-GlcNAc labeling. Although pathways through which glucose can feed into UDP-GlcNAc's glucosamine ring independent of the HBP have not been described in mammalian cells to our knowledge, we nevertheless tested the two major metabolic branch points diverging from UDP-GlcNAc, which mediate UDP-GalNAc and sialic acid synthesis. UDP-galactose-4-epimerase (GALE) interconverts UDP-GlcNAc and UDP-GalNAc, and UDP-GlcNAc-2-epimerase/ManAc kinase (GNE) initiates sialic acid biosynthesis. Silencing of neither GALE nor GNE reduced UDP-GlcNAc labeling from glucose, however, indicating that these enzymes are unlikely to facilitate a bypass pathway (*Figure 2—figure supplement 2D,E*). Although a minor con-tribution from another unknown pathway cannot be ruled out, the slight apparent differences in tim-ing of GlcNAc-P and UDP-GlcNAc labeling most likely reflect technical limitations. Regardless, the data clearly indicate that UDP-GlcNAc abundance is maintained despite reduced de novo hexos-amine synthesis from glucose (*Figure 2D*).

## GlcNAc salvage feeds UDP-GlcNAc pools in pancreatic cancer cells

As mentioned, UDP-GlcNAc can be generated via phosphorylation of free GlcNAc by NAGK-gener-ating GlcNAc-6-P (*Figure 2B*). When supplemented, GlcNAc is salvaged into the UDP-GlcNAc pool

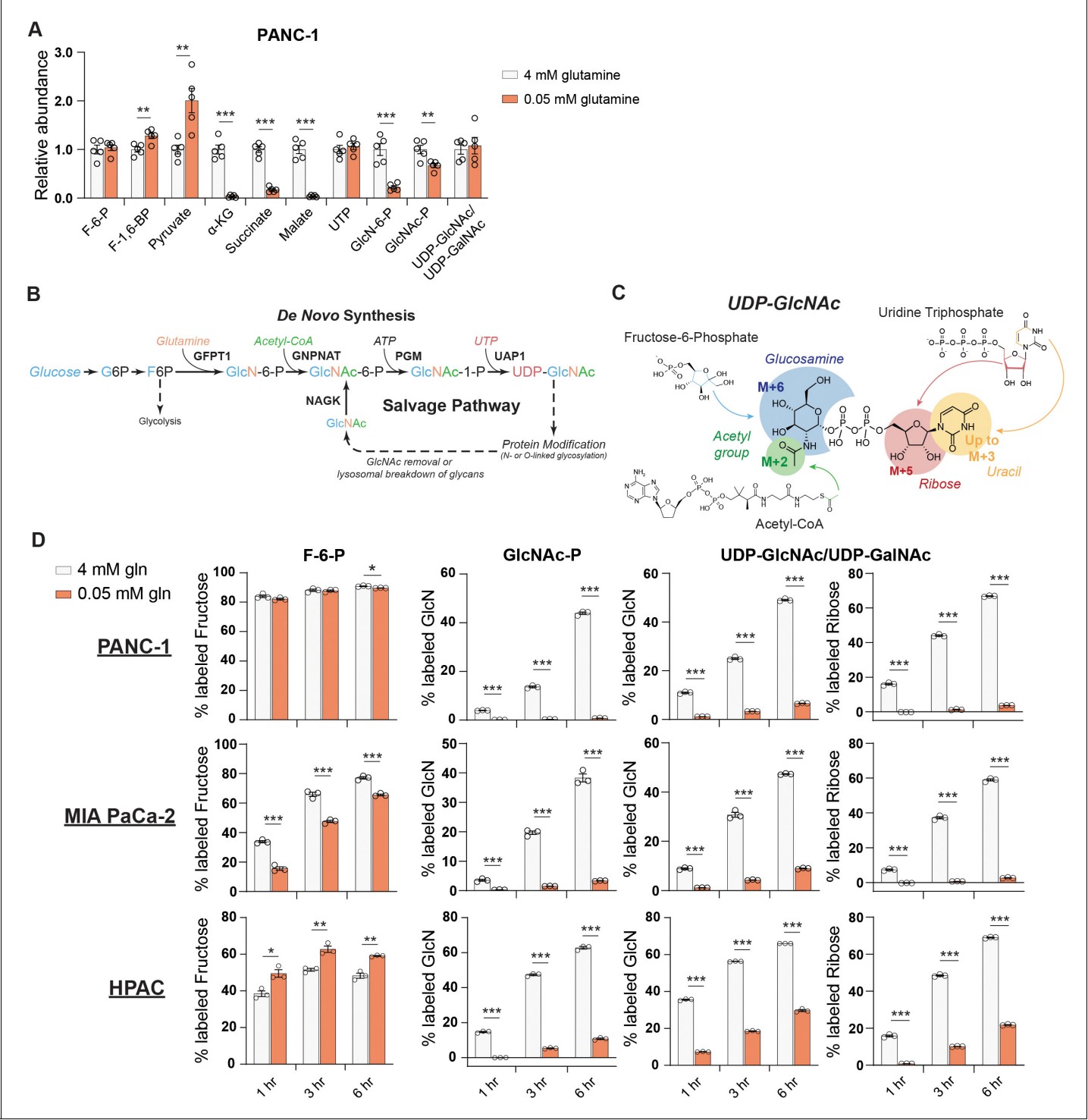

**Figure 2.** De novo UDP-GlcNAc synthesis is suppressed upon glutamine deprivation. (**A**) Metabolite measurements in PANC-1 cells after culture for 48 hr in 0.05 mM glutamine. Quantification is normalized to 4 mM glutamine condition. Statistical significance was calculated by unpaired t-test. Mean ± SEM of five biological replicates is represented. (**B**) Overview of the GlcNAc salvage pathway feeding into the HBP. GlcNAc scavenged from O-GlcNAc removal or lysosomal breakdown of glycans can be phosphorylated by NAGK and used to regenerate UDP-GlcNAc. (**C**) Overview of the incorporation of [13]C glucose into UDP-GlcNAc. Different parts of the molecule can be labeled from glucose-derived subunits; thus, isotopologs up to M+16 can be derived from glucose. (**D**) [13]C glucose tracing into F-6-P, GlcNAc-P, and UDP-GlcNAc in indicated glutamine concentrations. % labeled GlcN indicates sum of M+6 and M+8 isotopologs for GlcNAc-P and sum of M+6, M+8, M+11, and M+13 for UDP-GlcNAc. % labeled Ribose indicates

*Figure 2 continued on next page*

*Figure 2 continued*

sum of M+5, M+7, M+11, and M+13 for UDP-GlcNAc. All isotopologs are graphed in *Figure 2—figure supplement 1A*. Statistical significance was calculated by unpaired t-test. Mean ± SEM of three biological replicates is represented. *$p \leq 0.05$; **$p \leq 0.01$; ***$p \leq 0.001$.

The online version of this article includes the following figure supplement(s) for figure 2:

**Figure supplement 1.** De novo hexosamine synthesis is suppressed in low glutamine conditions.

**Figure supplement 2.** Uridine is salvaged in glutamine-restricted conditions.

**Figure supplement 3.** NAGK protein expression does not increase in low glutamine despite increase in *NAGK* gene expression.

(*Ryczko et al., 2016*; *Wellen et al., 2010*). Endogenous sources of GlcNAc may include removal of O-GlcNAc protein modifications or breakdown of glycoconjugates and extracellular matrix components. Notably, intracellular levels of GlcNAc increase upon glutamine restriction (*Figure 3A*). Yet, the significance of GlcNAc salvage to maintenance of UDP-GlcNAc pools has been little studied, and the proportion of UDP-GlcNAc generated via the NAGK-dependent salvage pathway is unknown.

*NAGK* mRNA expression increased in PDA cell lines in low glutamine conditions and in some cell lines also in low glucose (*Figure 2—figure supplement 3A,B*). *GFPT1* expression was also induced in both low glucose conditions, consistent with a prior report (*Moloughney et al., 2016*), and in low glutamine conditions (*Figure 2—figure supplement 3A*), even though de novo synthesis is suppressed when glutamine is limited. Protein levels of NAGK did not increase in concordance with mRNA at these time points, however, although a mobility shift potentially indicative of post-translational modification was apparent when protein lysates were run on a gel using a large electrophoresis system (see Materials and methods; *Figure 2—figure supplement 3C,D*). Removal of the phosphatase inhibitor $Na_3VO_4$ from the sample buffer prevented the mobility shift, suggesting that NAGK may be phosphorylated on one or more residues in low glutamine conditions (*Figure 2—figure supplement 3D*). Taken together, these data indicate that under low glutamine conditions, GlcNAc availability for salvage increases and the salvage enzyme NAGK is subject to regulation.

These findings prompted us to investigate the role of NAGK in UDP-GlcNAc synthesis in PDA cells. We functionally examined the role of NAGK in PDA cell lines by using CRISPR-Cas9 gene editing to generate NAGK knockout (KO) PANC-1 and MiaPaCa-2 clonal cell lines (*Figure 3—figure supplement 1A, B*). N-[1,2-$^{13}C_2$]acetyl-D-glucosamine ($^{13}$C GlcNAc) was efficiently salvaged in control cells, and this was suppressed by NAGK deletion, as evidenced by reduced fractional labeling of GlcNAc-P and UDP-GlcNAc (*Figure 3B*). Since we did not observe any residual protein expression, we hypothesized that the N-acetylgalactosamine (GalNAc) salvage enzyme GalNAc kinase (GALK2) might be responsible for the remaining GlcNAc salvage in the absence of NAGK. Indeed, silencing of GALK2 further suppressed incorporation of $^{13}$C GlcNAc into GlcNAc-P and UDP-GlcNAc in the NAGK KO cells (*Figure 3—figure supplement 1C*).

We hypothesized that knockout cells would conversely conduct increased de novo UDP-GlcNAc synthesis. To test this, we incubated cells with [U-$^{13}$C]-glucose and examined incorporation into GlcNAc-P and UDP-GlcNAc. Indeed, in the absence of NAGK, we observed increased glucose-dependent fractional labeling of the glucosamine ring of UDP-GlcNAc and GlcNAc-P, but not the ribose component of UDP-GlcNAc (*Figure 3C,D*, *Figure 3—figure supplement 1D,E*). This effect was also observed with knockdown of NAGK by shRNA, though to a lesser extent (*Figure 3—figure supplement 2A,B*). Incorporation of glucose into F-6-P did not change (*Figure 3C*), and the proportion of UDP-GlcNAc containing an M+5 ribose ring was also unchanged in knockout cells (*Figure 3D*), as expected. Thus, when NAGK is deleted and GlcNAc salvage is suppressed, de novo hexosamine synthesis increases.

We next assessed changes in the levels of hexosamine intermediates in control and NAGK KO cell lines. In PANC-1 KO cells in 4 mM glutamine, GlcN-P increased significantly, consistent with increased de novo synthesis in the absence of NAGK (*Figure 3E*). GlcNAc-P was modestly reduced in KO cells, though UDP-GlcNAc levels were maintained (*Figure 3E*). In MIA PaCa-2 cells, GlcNAc-P was markedly suppressed in the absence of NAGK, though UDP-GlcNAc was not (*Figure 3—figure supplement 2C*). We also measured HBP metabolites under glutamine restriction, where we expected NAGK would play a more significant role in UDP-GlcNAc generation. We were only able to measure metabolites accurately in PANC-1 KO cells because MIA PaCa-2 NAGK KO cells began

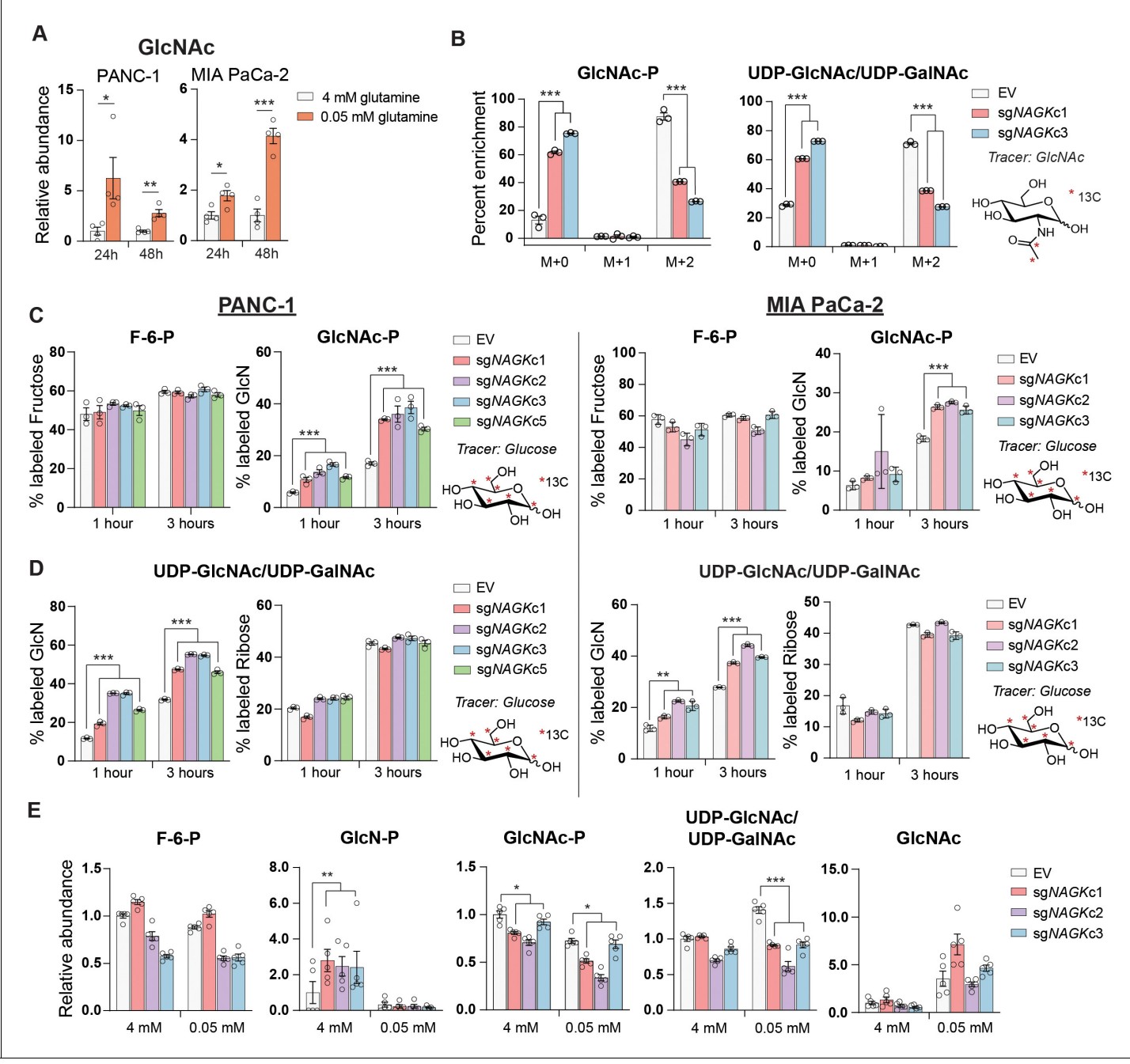

**Figure 3.** GlcNAc salvage feeds UDP-GlcNAc pools in pancreatic cancer cells. (**A**) Measurement of GlcNAc in PANC-1 and MIA PaCa-2 cells after incubation in the indicated concentrations of glutamine for 24 and 48 hr. Mean ± SEM of four biological replicates is represented. Statistical significance was calculated by unpaired t-test. (**B**) Measurement of 13C GlcNAc labeled on the acetyl group into GlcNAc-P and UDP-GlcNAc in NAGK knockout cells. Cells were incubated with 10 mM 13C GlcNAc for 6 hr. Mean ± SEM of three biological replicates is represented. Statistical significance was calculated by unpaired t-test comparing the mean incorporation of the two CRISPR clones and the empty vector (EV) control. (**C**) Labeling of F-6-P and GlcNAc-P from 13C glucose in PANC-1 (left) and MIA PaCa-2 (right) NAGK knockout cells. Statistical significance was calculated by unpaired t-test comparing the mean incorporation of the four CRISPR clones and the EV control. Mean ± SEM of three biological replicates is represented. (**D**) Percent of combined UDP-GlcNAc isotopologs containing a labeled glucosamine ring or a labeled ribose from UTP, calculated from S3.1 (**E**). Statistical significance was calculated by unpaired t-test comparing the mean incorporation of the four CRISPR clones and the EV control. Mean ± SEM of three biological replicates is represented. (**E**) Measurement of HBP metabolites in PANC-1 NAGK knockout cells cultured in the indicated concentrations of glutamine. Statistical significance was calculated by unpaired t-test comparing the mean incorporation of the four CRISPR clones and the EV control. Mean ± SEM of four biological replicates is represented. *p≤0.05; **p≤0.01; ***p≤0.01.

The online version of this article includes the following figure supplement(s) for figure 3:

*Figure 3 continued on next page*

Figure 3 continued
**Figure supplement 1.** NAGK knockout cells show increased enrichment of 13C glucose into hexosamine intermediates.
**Figure supplement 2.** HBP metabolites in NAGK KO cells.

to die quickly in low glutamine, which will be discussed further in the next section. GlcN-P decreased in control and KO cells, consistent with reduced de novo hexosamine synthesis (*Figure 3E*). GlcNAc-P levels decreased in low glutamine in control cells and decreased further in cells lacking NAGK, consistent with contributions from both de novo synthesis and salvage (*Figure 3E*). Reciprocally, GlcNAc abundance was elevated upon glutamine limitation in both control and NAGK KO cells (*Figure 3E*). UDP-GlcNAc abundance was modestly reduced in NAGK KO cells relative to controls under glutamine restriction, though levels were still comparable to that in high glutamine (*Figure 3E*), possibly reflecting changes in utilization. GALK2 silencing did not further suppress UDP-GlcNAc in NAGK KO cells (*Figure 3—figure supplement 2D*), suggesting that GALK2 may not have a major role in physiological GlcNAc salvage. Cumulatively, the data demonstrate that GlcNAc is salvaged into UDP-GlcNAc pools in PDA cells in a manner dependent at least in part on NAGK.

## NAGK knockout limits tumor growth in vivo

To test the role of NAGK in cell proliferation, we first monitored growth of NAGK KO cells compared to controls in 2D and 3D culture in 4 mM glutamine, finding minimal differences (*Figure 4A*, *Figure 4—figure supplement 1*). We hypothesized that NAGK KO cell proliferation would be impaired in 0.05 mM glutamine, where de novo UDP-GlcNAc synthesis is suppressed. Indeed, MIA PaCa-2 KO cells died more quickly in 0.05 mM glutamine than did control cells (*Figure 4A*). PANC-1 KO cells did not show this effect (*Figure 4A*), but we hypothesized that NAGK loss might have a stronger effect in vivo where tumor growth can be constrained by nutrient availability.

To gain initial insight into whether NAGK is likely to play a functional role in PDA progression in vivo, we queried publicly available datasets. From analysis of publicly available microarray data (*Pei et al., 2009*) and gene expression data from the Cancer Genome Atlas (TCGA), we indeed found *NAGK* expression to be increased in tumor tissue relative to adjacent normal regions of the pancreas or to pancreas GTEx data (*Figure 4B*, *Figure 4—figure supplement 1B*). *GFPT1* expression was also increased in tumor tissue (*Figure 4B*, *Figure 4—figure supplement 1B*), consistent with its regulation by mutant KRAS (*Ying et al., 2012*). Two other HBP genes, *PGM3* and *UAP1*, did not show significantly increased expression in PDA tumors in these datasets (*Figure 4B*, *Figure 4—figure supplement 1B*). We then studied the role of NAGK in tumor growth in vivo by injecting NAGK CRISPR KO cells into the flank of NCr nude mice. Final tumor volume and weight were markedly reduced in the absence of NAGK (*Figure 4C–D*). Of note, initial tumor growth was comparable between control and KO cells, but the NAGK knockout tumors either stopped growing or shrank while control tumors continued to grow larger (*Figure 4—figure supplement 1C*). Interestingly, KO tumor samples showed increased L-PHA signal (*Figure 4—figure supplement 1D*), indicating that NAGK deficiency results in altered glycosylation within tumors. This could possibly reflect either elevated de novo synthesis in the small tumors that form or differences in cellular composition. For example, activated fibroblast marker α-smooth muscle actin (α-SMA) was more abundant in the NAGK KO tumors (*Figure 4—figure supplement 1D*). Residual NAGK signal in whole tumors also presumably reflects expression in other cell types (*Figure 4—figure supplement 1D*), since NAGK was undetectable in the clonal cell lines used for injections (*Figure 3—figure supplement 1B*). Taken together, these data are consistent with the notion that NAGK is dispensable when nutrients are abundant but becomes more important as the tumors outgrow their original nutrient supply and become more dependent on scavenging and recycling and indicate that NAGK-mediated hexosamine salvage supports tumor growth in vivo.

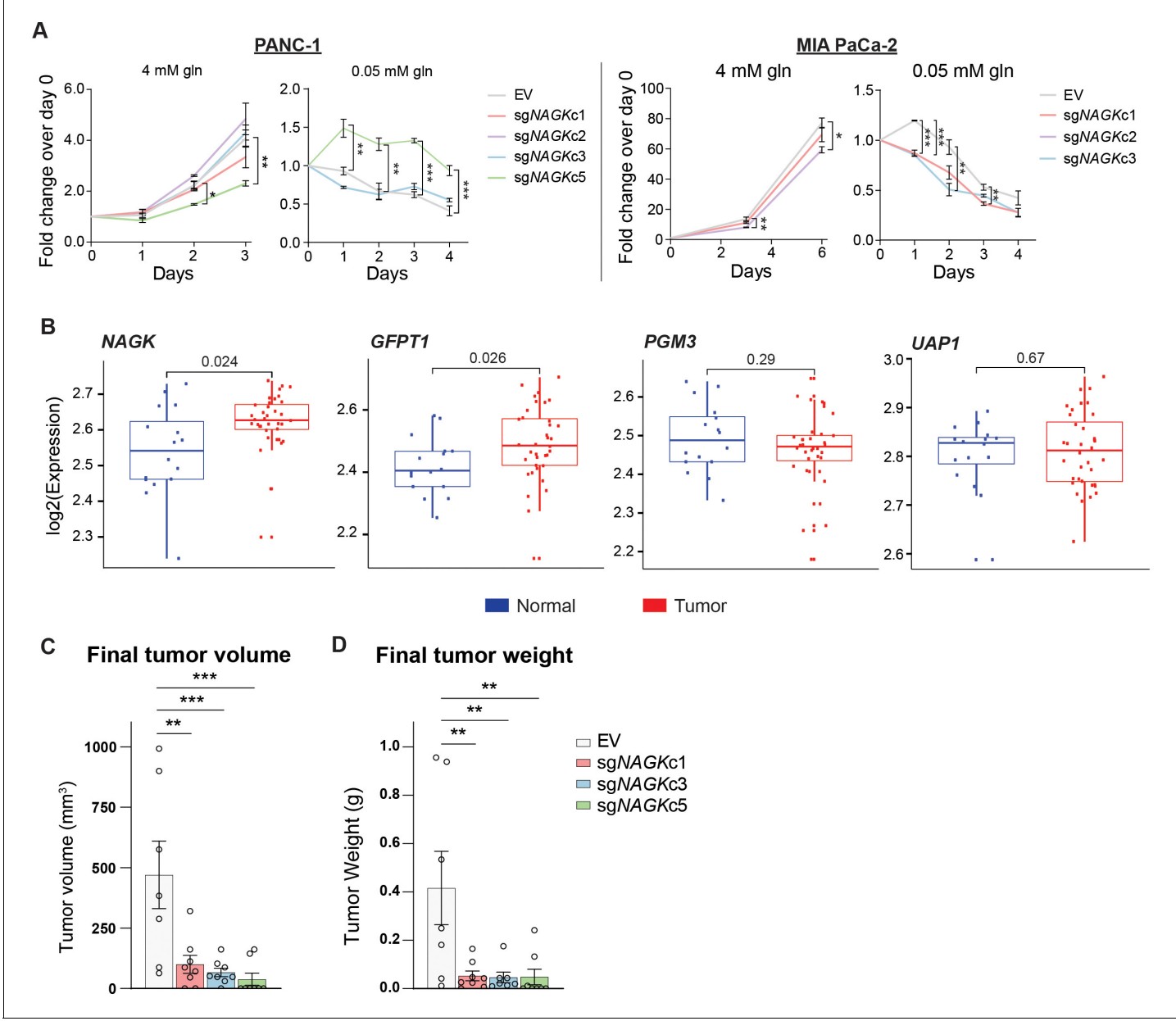

**Figure 4.** NAGK expression is increased in human PDA tumors and NAGK knockout reduces tumor growth in vivo. (**A**) 2D proliferation assay by cell count in PANC-1 and MIA PaCa-2 NAGK knockout cells. Mean ± SEM of three technical replicates is represented. Statistical significance was calculated using one-way ANOVA at each time point. (**B**) Gene expression data for NAGK, GFPT1, PGM3, and UAP1 in human PDA tumors compared with matched normal tissue. Statistical analysis was conducted by one-way ANOVA, and level of significance was defined as p≤0.01. (**C**) Final tumor volume and (**D**) final tumor weight of subcutaneous tumors generated from PANC-1 NAGK knockout cells in vivo. Cells were injected into the right flank of NCr nude mice, and tumor volume was calculated from caliper measurements. Statistical significance was calculated by one-way ANOVA comparing each mean to the EV control mean. Mean ± SEM of biological replicates is represented (n = 8 each group). *p ≤ 0.05; **p ≤ 0.01; ***p ≤ 0.001.

The online version of this article includes the following figure supplement(s) for figure 4:

**Figure supplement 1.** NAGK knockout tumors show increased L-PHA binding.

# Discussion

In this study, we identify a key role for NAGK in salvaging GlcNAc for UDP-GlcNAc synthesis in PDA cells. We show that glutamine deprivation suppresses de novo hexosamine biosynthesis, which is reciprocally increased upon NAGK deletion. Glutamine deprivation also results in increased availability of GlcNAc for salvage. *NAGK* expression is elevated in human PDA tumors, and NAGK deficiency suppresses GlcNAc salvage in cells and tumor growth in mice.

This work raises several key questions for future investigation. First, the sources of GlcNAc salvaged by NAGK remain to be fully elucidated. GlcNAc may be derived from recycling of GlcNAc following O-GlcNAc removal or breakdown of glycoconjugates. Additionally, GlcNAc may be recovered from the environment. Nutrient scavenging via macropinocytosis is a key feature of PDA (*Commisso et al., 2013*; *Kamphorst et al., 2015*). Macropinocytosis has mostly been associated with scavenging of protein to recover amino acids, but lysosomal break down of glycoproteins may also release sugars including GlcNAc. Additionally, ECM components, including hyaluronic acid (HA), which is a polymer of GlcNAc and glucuronic acid disaccharide units, may be additional sources of GlcNAc for salvage in the tumor microenvironment. Indeed, in a manuscript co-submitted with this one, *Kim et al., 2020* identify HA as a major source of scavenged GlcNAc . Our manuscript and the Kimet al.'s manuscript together indicate that NAGK may take on a heightened importance in the context of high GlcNAc availability and nutrient deprivation, a situation that is likely to occur within the tumor microenvironment.

Furthermore, the key fates of UDP-GlcNAc that support tumor growth remain to be elucidated. Sufficient UDP-GlcNAc is required for protein glycosylation to maintain homeostasis and prevent ER stress, particularly in a rapidly dividing cell. Additionally, a wide range of cancers exhibit elevated O-GlcNAc, which could contribute to driving pro-tumorigenic transcriptional and signaling programs. In PDA specifically, the glycan CA19-9 is currently used as a biomarker for disease progression and recent studies point to a functional role for CA19-9 in tumorigenesis (*Engle et al., 2019*). UDP-GlcNAc is also required for HA synthesis, which is present in low amounts in normal pancreas but increases in PanIN lesions and PDA (*Provenzano et al., 2012*). PDA cells are capable of producing HA in vitro (*Mahlbacher et al., 1992*). Depletion of fibroblasts in an autochthonous PDA mouse model results in a decrease in collagen I, but not HA in the tumor microenvironment, indicating that HA must be generated by another cell type, possibly the tumor cells themselves (*Özdemir et al., 2014*). Previous studies demonstrated that treatment of PDA with exogenous hyaluronidase can increase vascularization and improve drug delivery to the tumor (*Jacobetz et al., 2013*; *Provenzano et al., 2012*), although a phase III clinical trial reported no improvement in overall patient survival when combining pegylated hyaluronidase with nab-paclitaxel plus gemcitabine (*Van Cutsem et al., 2020*). Recently, it was shown that inhibiting the HBP by treatment with DON depletes HA and collagen in an orthotopic mouse model. DON treatment also increased CD8 T-cell infiltration into the tumor, sensitizing the tumor to anti-PD1 therapy (*Sharma et al., 2020*). Thus, targeting the HBP holds promise for improving the efficacy of other therapeutics. The findings of the current study suggest that in addition to de novo hexosamine synthesis, targeting of hexosamine salvage warrants further investigation in terms of potential for therapeutic intervention. Of note, an inhibitor targeting PGM3, which converts GlcNAc-6-P to GlcNAc-1-P and is thus required for both de novo UDP-GlcNAc synthesis and GlcNAc recycling, showed efficacy in treating gemcitabine-resistant patient-derived xenograft PDA models (*Ricciardiello et al., 2020*), as well as in breast cancer xenografts (*Ricciardiello et al., 2018*).

Finally, almost nothing is currently known about the role of NAGK and GlcNAc salvage in normal physiology. Even in non-cancerous IL-3-dependent hematopoietic cells, a substantial proportion of the UDP-GlcNAc pool remains unlabeled from $^{13}$C-glucose (*Wellen et al., 2010*), suggesting that salvage may contribute to UDP-GlcNAc pools in a variety of cell types. However, while GFPT1 is required for embryonic development in mice, NAGK knockout mouse embryos are viable (*Dickinson et al., 2016*). NAGK deficiency has not yet been characterized in postnatal or adult mice. Perhaps GlcNAc salvage is dispensable when nutrients are available and cells are not dividing, as in most healthy tissues. However, in a tumor, in which cells are proliferating and nutrients are spread thin, NAGK and GlcNAc salvage may become more important in feeding UDP-GlcNAc pools. Related questions include elucidating the mechanisms regulating NAGK gene expression and

putative post-translational modification, as well as understanding the role of GALK2 in hexosamine salvage.

In sum, we report a key role for NAGK in feeding UDP-GlcNAc pools in PDA cells and in supporting xenograft tumor growth. Further investigation will be needed to elucidate the physiological functions of NAGK, as well as the mechanisms through which it supports tumor growth and its potential role in modulating therapeutic responses.

# Materials and methods

## Key resources table

| Reagent type (species) or resource | Designation | Source or reference | Identifiers | Additional information |
|---|---|---|---|---|
| Cell line (*Homo sapiens*) | MIA PaCa-2 | ATCC | CRL-1420 (RRID:CVCL_0428) | |
| Cell line (*Homo sapiens*) | PANC-1 | ATCC | CRL-1469 (RRID:CVCL_0480) | |
| Cell line (*Homo sapiens*) | HPAC | ATCC | CRL-2119 (RRID:CVCL_3517) | |
| Cell line (*Homo sapiens*) | AsPC-1 | ATCC | CRL-1682 (RRID:CVCL_0152) | |
| Cell line (*Homo sapiens*) | BxPC-3 | ATCC | CRL-1687 (RRID:CVCL_0186) | |
| Cell line (*Homo sapiens*) | HCT 116 | ATCC | CCL-247 (RRID:CVCL_0291) | |
| Cell line (*Homo sapiens*) | SW480 | ATCC | CCL-228 (RRID:CVCL_0546) | |
| Recombinant DNA reagent | lentiCRISPRv2 | Addgene | 98290 (RRID:Addgene_98290) | |
| Antibody | O-GlcNAc CTD1106, mouse monoclonal | Cell Signaling | 9875S (RRID:AB_10950973) | WB (1:1000) |
| Antibody | Tubulin, mouse monoclonal | Sigma | T6199 (RRID:AB_477583) | WB (1:1000) |
| Antibody | HSP60, rabbit monoclonal | Cell Signaling | 12165S (RRID:AB_2636980) | WB (1:1000) |
| Antibody | NAGK, rabbit polyclonal | Atlas Antibodies | HPA035207 (RRID:AB_10602031) | WB (0.4 ug/mL) |
| Antibody | NAGK, rabbit polyclonal | Proteintech | 15051–1-AP (RRID:AB_2152368) | WB (1:1000) |
| Antibody | AKT phosphoS473, rabbit monoclonal | Cell Signaling | 4060 (RRID:AB_2315049) | WB (1:1000) |
| Antibody | Viniculin, mouse monoclonal | Sigma | V9264 (RRID:AB_10603627) | WB (1:10000) |
| Chemical compound | [U-13C]-glucose | Cambridge Isotopes | CLM-1396–1 | |
| Chemical compound | 13C GlcNAc | Omicron Biochemicals | GLC-006 | |
| Strain, strain background (*Mus musculus*) | NCr nude mice | Taconic | CrTac:NCr-Foxn1nu (RRID:IMSR_TAC:ncrnu) | |
| Sequence-based reagent | *GFPT1* forward | This paper | RT-qPCR primers | CTCTGGCTTTGGTGGATAAA |
| Sequence-based reagent | *GFPT1* reverse | This paper | RT-qPCR primers | GCAACCACTTGCTGAAGA |
| Sequence-based reagent | *NAGK* forward | This paper | RT-qPCR primers | GTGCTCATATCTGGAACAGG |
| Sequence-based reagent | *NAGK* reverse | This paper | RT-qPCR primers | ACCCTCATCACCCATCATA |
| Sequence-based reagent | *HPRT* forward | This paper | RT-qPCR primers | ATTATGCCGAGGATTTGGAA |

*Continued on next page*

*Continued*

| Reagent type (species) or resource | Designation | Source or reference | Identifiers | Additional information |
|---|---|---|---|---|
| Sequence-based reagent | *HPRT* reverse | This paper | RT-qPCR primers | CCCATCTCCTTCATGACATCT |
| Sequence-based reagent | *RPL19* forward | This paper | RT-qPCR primers | CAAGAAGGAGGAGATCATCAAG |
| Sequence-based reagent | *RPL19* reverse | This paper | RT-qPCR primers | ATCACAGAGGCCAGTATGTA |
| Sequence-based reagent | sg*MGAT5* mouse forward | *Doench et al., 2016* | CRISPR deletion primers | CACCGGCTGTCATGACACCAGCGTA |
| Sequence-based reagent | sg*MGAT5* mouse reverse | *Doench et al., 2016* | CRISPR deletion primers | AAACTACGCTGGTGTCATGACAGCC |
| Sequence-based reagent | sg*NAGK*#1 forward | *Doench et al., 2016* | CRISPR deletion primers | CACCGTTGACGTAGCCGATATCATG |
| Sequence-based reagent | sg*NAGK*#1 reverse | *Doench et al., 2016* | CRISPR deletion primers | AAACCATGATATCGGCTACGTCAAC |
| Sequence-based reagent | sg*NAGK*#2 forward | *Doench et al., 2016* | CRISPR deletion primers | CACCGTGCTTGGTGTGCGATCCAGT |
| Sequence-based reagent | sg*NAGK*#2 reverse | *Doench et al., 2016* | CRISPR deletion primers | AAACACTGGATCGCACACCAAGCAC |
| Sequence-based reagent | sg*NAGK*#3 forward | *Doench et al., 2016* | CRISPR deletion primers | CACCGCTCTACACCCCCATAGATCG |
| Sequence-based reagent | sg*NAGK*#3 reverse | *Doench et al., 2016* | CRISPR deletion primers | AAACCGATCTATGGGGGTGTAGAGC |
| Transfected construct (human) | siRNA non-targeting control | Santa Cruz Biotechnology | SC-37007 | |
| Transfected construct (human) | siGALE | Santa Cruz Biotechnology | SC-78950 | |
| Transfected construct (human) | siGNE | Santa Cruz Biotechnology | SC-60693 | |
| Transfected construct (human) | siGALK2 | Santa Cruz Biotechnology | SC-90002 | |

## Cell culture

Cells were cultured in DMEM high glucose (Gibco, 11965084) with 10% calf serum (Gemini GemCell U.S. Origin Super Calf Serum, 100–510), unless otherwise noted. Glucose- or glutamine-restricted media was prepared using glucose, glutamine, and phenol red-free DMEM (Gibco, A1443001) supplemented with glucose (Sigma-Aldrich, G8769), glutamine (Gibco, 25030081), and dialyzed fetal bovine serum (Gemini, 100–108). For all glutamine restriction experiments except S2.3 D, cells were plated 2–3× more densely for the nutrient-restricted condition samples to achieve similar confluency at the experiment endpoint. One percent oxygen levels were achieved by culturing cells in a Whitley H35 Hypoxystation (Don Whitley Scientific). ATCC names and numbers for the cell lines used in this study are as follows: MIA PaCa-2 (ATCC# CRL-1420), PANC-1 (ATCC# CRL-1469), HPAC (ATCC# CRL-2119), AsPC-1 (ATCC# CRL-1682), BxPC-3 (ATCC# CRL-1687), HCT 116 (ATCC# CCL-247), and SW480 (ATCC# CCL-228). All cells were routinely tested for mycoplasma and authenticated by short tandem repeat (STR) profiling using the GenePrint 10 System (Promega, B9510).

Generation of CRISPR cell lines sgRNA sequences targeting *NAGK* or *Mgat5* from the Brunello and Brie libraries (*Doench et al., 2016*) was cloned into the lentiCRISPRv2 vector (*Sanjana et al., 2014*). Lentivirus was produced in 293 T cells according to standard protocol. Cells were then infected with the CRISPR lentivirus and selected with puromycin. Cells were plated at very low density into 96-well plates to establish colonies generated from single-cell clones. *Mgat5* gene disruption was validated by qPCR and L-PHA binding. *NAGK* gene disruption was validated by qPCR, western blot, and $^{13}$C-GlcNAc tracing. Seven *NAGK* knockout clonal cell lines established from three

different sgRNAs, four in PANC-1 cells and three in MIA PaCa-2 cells, were chosen for use in the study. Please see table at end of methods for primer sequences of guides used.

## Western blotting

For protein extraction from cells, cells were kept on ice and washed three times with PBS, then scraped into PBS and spun down at 200 g for 5 min. The cell pellet was resuspended in 50–100 μL RIPA buffer (1% NP-40, 0.5% deoxycholate, 0.1% SDS, 150 mM NaCl, 50 mM Tris plus protease inhibitor cocktail [Sigma-Aldrich, P8340] and phosSTOP [Sigma-Aldrich, 04906845001]), and lysis was allowed to continue on ice for 10 min. Cells were sonicated with a Fisherbrand Model 120 Sonic Dismembrator (Fisher Scientific, FB120A110) for three pulses of 20 s each at 20% amplitude. Cell lysate was spun down at 15,000 g for 10 min at 4℃, and supernatant was transferred to a new tube. For protein extraction from tissue, the sample was resuspended in 500 μL RIPA buffer and homogenized using a TissueLyser (Qiagen, 85210) twice for 30 s at 20 Hz. Following incubation on ice for 10 min, the same procedure was followed as for cells. For both cells and tissue, lysate samples were stored at −80℃ until analysis by immunoblot. All blots were developed using a LI-COR Odyssey CLx system. Antibodies used in this study were as follows: O-GlcNAc CTD110.6 (Cell Signaling 9875S), tubulin (Sigma T6199), HSP60 (Cell Signaling 12165S), NAGK (Atlas Antibodies, HPA035207), and PARP (Cell Signaling 9532).

For blots showing the mobility shift for NAGK in low glutamine, samples were prepared in lysis buffer containing 50 mM Tris pH 8.0, 150 mM NaCl, 0.5% IGEPAL CA-630 (Sigma, I3021), 1 mM PMSF, 1.5 μM aprotinin, 84 μM leupeptin, 1 μM pepstatin A, ±10 mM NaF, and 20 mM $Na_3VO_4$ as indicated. To visualize the NAGK mobility shift in response to low glutamine, 20 μg total protein per sample was separated across 12.5 cm of 11% SDS-PAGE resolving space under reducing conditions using the large electrophoresis systems available from C.B.S. Scientific until approximately 3 cm of separation was obtained between the 25 and 37 kDa protein standards (Bio-Rad; 1610375). Using electrophoresis, proteins were transferred (30 V, 4℃, overnight) to 0.45 μM pore size nitrocellulose membrane (Amersham, 10600002). The primary antibodies used were NAGK (Proteintech, 15051–1-AP), AKT phosphoS473 (Cell Signaling Technology, 4060), and Vinculin (Sigma, V9264). Membranes were developed using the LI-COR Odyssey CLx system.

## RT-qPCR

For RNA extraction from cells, cells were put on ice, washed with PBS, and scraped into PBS. Samples were then spun down at 200 g for 5 min and resuspended in 100 μL Trizol (Life Technologies). For RNA extraction from tissue, samples were resuspended in 500 μL Trizol and homogenized using a TissueLyser twice for 30 s at 20 Hz. For both cells and tissue, RNA was extracted following the Trizol manufacturer protocol. cDNA was prepared using high-capacity RNA-to-cDNA master mix (Applied Biosystems, 4368814) according to kit instructions. cDNA was diluted 1:20 and amplified with PowerUp SYBR Green Master Mix (Applied Biosystems, A25778) using a ViiA-7 Real-Time PCR system. Fold change in expression was calculated by the $\Delta\Delta C_t$ method using HPRT as a control. Please see table at end of chapter for primer sequences.

## Lectin binding assay

Cells were put on ice, washed with PBS, and then scraped into PBS. Samples were then spun down at 200 g for 5 min and resuspended in 3% BSA with fluorophore-conjugated lectin added 1:1000 (Vector Labs FL-1111–2). Samples were covered and incubated on ice for 30 min at room temperature, then spun down, and resuspended in PBS before analysis with an Attune NxT Flow Cytometer (Thermo Fisher Scientific). Data was further analyzed using FlowJo 8.7.

## Metabolite quantitation and labeling

For all metabolite quantitation experiments, each sample was collected from a 10 cm sub-confluent plate of cells. To achieve similar confluency and protein content at the experiment end point, cells were initially plated more densely for the nutrient-deprived samples than for the nutrient-replete samples. For low glutamine experiments, PANC-1 cells were plated $3 \times 10^5$ for 4 mM glutamine samples and $5.5 \times 10^5$ for 0.05 mM samples. MIA PaCa-2 cells were plated $3 \times 10^5$ for 4 mM samples and $1.2 \times 10^6$ for 0.05 mM samples.

Samples were prepared according to *Guo et al., 2016b*. Briefly, cells were put on ice and washed 3× with PBS. Then, 1 mL of ice cold 80% methanol was added to the plate, and cells were scraped into solvent and transferred to a 1.5 mL tube. For quantitation experiments, internal standard containing a mix of $^{13}$C labeled metabolites was added at this time. Samples were then sonicated and spun down, and the supernatants were dried down under nitrogen. The dried samples were then resuspended in 100 µL of 5% sulfosalicylic acid and analyzed by liquid chromatography–high-resolution mass spectrometry as reported (*Guo et al., 2016b*) with the only modification that the LC was coupled to a Q Exactive-HF with a heated ESI source operating in negative-ion mode alternating full scan and MS/MS modes. The [M-H]$^-$ ion of each analyte and its internal standard was quantified, with peak confirmation by MS/MS. GlcNAc quantification was done on a triple quadropole instrument exactly as described (*Guo et al., 2016b*). Data analysis was conducted in Thermo XCalibur 3.0 Quan Browser and FluxFix (*Trefely et al., 2016*). For quantitation experiments, samples were normalized first to peak integrations of $^{13}$C-labeled internal standard components and then to protein content in the sample, measured by BCA assay. Relative quantification was then calculated by normalizing to the control condition in each experiment.

For glucose labeling experiments, cells were cultured in DMEM without glucose, glutamine, or phenol red supplemented with 10 mM [U-$^{13}$C]-glucose (Cambridge Isotopes, CLM-1396–1), 4 mM glutamine, and 10% dialyzed fetal bovine serum. Cells were incubated for the indicated time, and samples were prepared as above. For GlcNAc labeling experiments, cells were cultured in DMEM without glucose, glutamine, or phenol red supplemented with 10 mM N-[1,2–13 C2]acetyl-D-glucosamine ($^{13}$C GlcNAc) (Omicron Biochemicals, GLC-006), 4 mM glutamine, 10 mM glucose, and 10% dialyzed fetal bovine serum. Cells were incubated for the indicated time, and samples were prepared as above.

## Soft agar colony formation assay

Cells were trypsinized and counted using a Bright-Line hemacytometer (Sigma, Z359629). The bottom agar layer was prepared by adding Bacto Agar (BD Bioscience, 214050) to cell culture media for a final concentration of 0.6%. Two milliliter bottom agar was added to each well of a six-well tissue culture plate. Once bottom agar solidified, top layer agar was prepared by combining trypsinized cells with the bottom agar mix for a final concentration of 0.3% Bacto Agar. One milliliter top layer agar was added to each well with a bottom layer of agar. Cells were plated $2.5 \times 10^4$ per well. 0.5 mL DMEM high glucose with 10% calf serum was added to cells every 7 days. Images were taken after 3 weeks. Images were blinded, and colonies per image were counted using ImageJ (*Schneider et al., 2012*).

## 2D proliferation assay

Cells were plated $3.5 \times 10^4$ per well of a six-well plate. For each day that counts were recorded, three wells were trypsinized and cells were counted twice using a hemocytometer (Sigma, Z359629). The average of the two counts was recorded for each well, and the average count of the three wells was used to graph the data. For proliferation assays in 0.05 mM glutamine, trypan blue was used during cell counts.

## Bioinformatics data analysis

The PDAC expression profiling dataset (GEO accession GSE16515, *Pei et al., 2009*) from NCBI GEO Profile database (*Edgar et al., 2002*) was used to compare the expression level between human normal and PDAC tumor samples. The dataset consists of 52 samples, in which 16 samples are matched tumor and normal tissues, and 20 samples are only tumor tissues. The statistical analysis was conducted by one-way ANOVA; the level of significance was evaluated by $p < 0.01$ and plotted in box-and-whisker diagram. Comparison of HBP gene expression between tumor (TCGA PAAD dataset) and normal tissue (GTEx) was also conducted using GEPIA2 (*Tang et al., 2019*).

## Tumor growth in vivo

$3 \times 10^6$ PANC-1 NAGK CRISPR cells were injected with 1:1 Matrigel (Corning, CB354248) into the flanks of NCr nude mice and measured with calipers once per week for 22 weeks. At the experiment end point (22 weeks or when tumor reached 20 mm in length), mice were euthanized with $CO_2$ and

cervical dislocation. Tumors were removed, weighed, cut into pieces for analysis, and frozen. All animal experiments were approved by the University of Pennsylvania and the Institutional Animal Care and Use Committee (IACUC).

## Acknowledgements

Funding sources: This work was supported by R01CA174761 and R01CA228339 to KEW. This work was also funded in part under a grant with the Pennsylvania Department of Health to KEW and IAB. The Department specifically disclaims responsibility for any analyses, interpretations, or conclusions. IAB acknowledges support of NIH Grants P30ES013508 and P30CA016520. JB acknowledges support of NIH Grant R01CA046595. SLC received support from T32CA115299 and F31CA217070, as well as from a Patel Family Scholar Award. HA was supported by post-doctoral fellowship K00CA212455. TT is supported by the National Cancer Institute through pre-doctoral fellowship F31CA243294 and acknowledges the Blavatnik Family for a predoctoral fellowship. LI is supported by T32 GM-07229 and T32 CA115299. ST is supported by the American Diabetes Association through post-doctoral fellowship 1–18-PDF-144. Funding sources were not involved in study design, data collection and interpretation, or the decision to submit the work for publication.

## Additional information

### Competing interests

Ian A Blair: IAB is a founder of Proteoform Bio and a paid consultant for Calico, Chimerix, PTC Therapeutics, Takeda Pharmaceuticals, and Vivo Capital. The other authors declare that no competing interests exist.

### Funding

| Funder | Grant reference number | Author |
| --- | --- | --- |
| National Cancer Institute | R01CA174761 | Kathryn E Wellen |
| National Cancer Institute | R01CA228339 | Kathryn E Wellen |
| Pennsylvania Department of Health | | Ian A Blair<br>Kathryn E Wellen |
| National Institute for Environmental Studies | P30ES013508 | Ian A Blair |
| National Cancer Institute | P30CA016520 | Ian A Blair |
| National Cancer Institute | R01CA046595 | John Blenis |
| National Cancer Institute | T32CA115299 | Sydney Campbell<br>Luke Izzo |
| National Cancer Institute | F31CA217070 | Sydney Campbell |
| National Cancer Institute | K00CA212455 | Hayley Affronti |
| National Cancer Institute | F31CA243294 | Tiffany Tsang |
| National Institute of General Medical Sciences | T32GM07229 | Luke Izzo |
| American Diabetes Association | 1-18-PDF-144 | Sophie Trefely |

The funders had no role in study design, data collection and interpretation, or the decision to submit the work for publication.

### Author contributions

Sydney Campbell, Conceptualization, Data curation, Formal analysis, Funding acquisition, Validation, Investigation, Visualization, Methodology, Writing - original draft, Writing - review and editing; Clementina Mesaros, Formal analysis, Investigation, Methodology, Writing - review and editing; Luke Izzo, Hayley Affronti, Michael Noji, Bethany E Schaffer, Tiffany Tsang, Salisa Kruijning,

Investigation, Writing - review and editing; Kathryn Sun, Formal analysis; Sophie Trefely, Formal analysis, Investigation, Writing - review and editing; John Blenis, Funding acquisition, Writing - review and editing; Ian A Blair, Supervision, Funding acquisition, Writing - review and editing; Kathryn E Wellen, Conceptualization, Formal analysis, Supervision, Funding acquisition, Methodology, Writing - original draft, Writing - review and editing

### Author ORCIDs
Sydney Campbell  https://orcid.org/0000-0002-7861-5609
Michael Noji  http://orcid.org/0000-0002-6996-5367
Kathryn E Wellen  https://orcid.org/0000-0002-2281-0042

### Ethics
Animal experimentation: This study was performed in strict accordance with the recommendations in the Guide for the Care and Use of Laboratory Animals of the National Institutes of Health. All of the animals were handled according to approved institutional animal care and use committee (IACUC) protocols (#805142) of the University of Pennsylvania.

### Decision letter and Author response
Decision letter https://doi.org/10.7554/eLife.62644.sa1
Author response https://doi.org/10.7554/eLife.62644.sa2

## Additional files
### Supplementary files
• Transparent reporting form

### Data availability
All data generated or analysed during this study are included in the manuscript and supporting files. Raw data files have been provided for all western blots.

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
