## [Decision Letter]

**Acceptance summary:**

In this manuscript, the authors investigate mechanisms by which pancreatic cancer cells maintain metabolite pools in the context of nutrient scarcity. How tumors, in particular notoriously-nutrient deprived tumors found in the pancreas, sustain survival and proliferation in the face of nutrient limitation is an area of profound interest to the fields of cancer and metabolism. Here, the authors identify a hexosamine salvage pathway that enables pancreatic cancer cells to maintain precursors required for protein glycosylation even in the context of severe nutrient restriction. The authors find that glutamine restriction activates N-acetylglucosamine salvage mediated by the enzyme N-acetylglucosamine kinase (NAGK) and identify NAGK as a critical mediator of pancreatic cancer cell growth in vivo. This work reveals the interplay between de novo synthesis and salvage of metabolites required for protein glycosylation and provides insight into the mechanisms underpinning the metabolic plasticity and resilience of pancreatic tumors.

**Decision letter after peer review:**

Thank you for submitting your article "Glutamine deprivation triggers NAGK-dependent hexosamine salvage" for consideration by *eLife*. Your article has been reviewed by 3 peer reviewers, one of whom is a member of our Board of Reviewing Editors, and the evaluation has been overseen by Richard White as the Senior Editor. The reviewers have opted to remain anonymous.

The reviewers have discussed the reviews with one another and the Reviewing Editor has drafted this decision to help you prepare a revised submission.

As the editors have judged that your manuscript is of interest, but as described below that additional experiments are required before it is published, we would like to draw your attention to changes in our revision policy that we have made in response to COVID-19 (https://elifesciences.org/articles/57162). First, because many researchers have temporarily lost access to the labs, we will give authors as much time as they need to submit revised manuscripts. We are also offering, if you choose, to post the manuscript to bioRxiv (if it is not already there) along with this decision letter and a formal designation that the manuscript is "in revision at eLife". Please let us know if you would like to pursue this option. (If your work is more suitable for medRxiv, you will need to post the preprint yourself, as the mechanisms for us to do so are still in development.)

Summary:

In this manuscript, the authors investigate mechanisms by which pancreatic cancer cells maintain UDP-GlcNAc pools for protein glycosylation within a nutrient-poor microenvironment. This work gains significance in light of recent publications documenting the importance of protein glycosylation for PDAC progression (i.e., PMID: 30853214, PMID: 29100056) and provides evidence in support of a novel mechanism by which pancreatic cancer cells maintain metabolite pools in the context of nutrient scarcity. The authors report the surprising observation that cultured pancreatic cancer cells appear to preserve protein glycosylation despite reduced glutamine availability. The proposed mechanism implicates NAGK as a mediator of GlcNAc salvage for UDP-GlcNAc synthesis under glutamine deprivation, which both suppresses de novo UDP-GlcNAc synthesis through the hexosamine biosynthesis pathway and induces NAGK expression. NAGK has been the subject of very little study in cancer and is entirely uncharacterized in pancreatic cancer to date, underscoring the novelty of this work and its likelihood to open up new lines of investigation in PDAC and other cancer types. Though the authors' focus on glutamine is understandable, experiments under glucose-limiting conditions are also performed here given the requirement for both nutrients to initiate the hexosamine biosynthetic pathway, and glucose restriction is relevant to the pancreatic tumor microenvironment (PMID: 30990168). Further examination of the significance of low-glucose conditions for NAGK activity and UDP-GlcNAc maintenance in PDAC cells would be helpful to place these findings in context. The authors should additionally address the following points to clarify their model and strengthen their conclusions:

Essential revisions:

1. The reviewers expressed some confusion with isotope tracing data presented in Figure 2 and Figure S2. Figure S2 shows no apparent M+6 in GlcNAc-P (in contrast to what is shown in Figure 2) and a similar amount of M+6 enrichment in UDP-GlcNAc in both high and low glutamine conditions, contrasting with reduced M+6 following glutamine deprivation in Figure 2C. Is it possible that Figure 2C might be showing M+8 for GlcNAc-P (middle) and M+5 for UTP (bottom), instead of what is listed? A related question based on Figure 2C is why, under glutamine restriction, GlcNAc-P labeling is completely abolished but labeling of UDP-GlcNAc (which is downstream) is only decreased by about half?

Given the potential issues with these data, it is hard to evaluate the conclusion that glucose tracing to UDP-GlcNAc is reduced during conditions of low glutamine. Importantly, Figure S2C appears to show that both M+6 and M+8 of UDP-GlcNAc is maintained during low glutamine, raising the possibility that synthesis of the glucosamine ring is not disrupted by low glutamine, which appears based on Figure S2 to specifically reduce synthesis of UTP. Given the critical importance of these tracing data to the central model of the paper-namely, that glutamine restriction reduces de novo GlcNAc synthesis-it is important for the authors to address these concerns. Additional concerns raised by the isotope tracing data are listed below.

2. In Figure 1, the authors show that protein glycosylation is preserved in PDAC cells even under low glutamine conditions. Given that PDAC cells are reported to be quite reliant on glutamine availability, is the viability of these cells affected under these growth conditions? Can the authors design experiments to determine whether the modification is maintained because of sustained deposition of GlcNAc or because of changes in cell survival/protein turnover and/or removal of glycoslyation?

3. It will be important for the authors to investigate whether NAGK deletion affects levels of glycosylation in cultured cells (especially during culture under low glutamine) and, if possible, in tumor samples. Figure 3D shows that UDP-GlcNAc pools are equivalent in NAGK deleted cells cultured in low or high glutamine, which at face value argues against NAGK as an important source of UDP-GlcNAc during glutamine restriction, but this interpretation could be affected by changes in overall cellular capacity for protein glycosylation.

4. As NAGK loss-of-function experiments are limited to the PANC-1 cell line, the authors should repeat a few key experiments in an additional NAGK knockdown/knockout PDAC cell line.

5. The authors nicely demonstrate that glutamine deprivation reduces flux through the HBP while inducing NAGK and enabling UDP-GlcNAc maintenance via GlcNAc salvage. The PDAC microenvironment also features restricted levels of glucose, such that PDAC cells face reduced levels of both nutrients relevant to the HBP in intact tumors. However, the impact of glucose restriction on UDP-GlcNAc pools presented in Figure S2B is somewhat difficult to reconcile with the compelling results of in vivo studies presented in Figure 4E and F, which suggest a critical role for NAGK to support PDAC cell proliferation within a nutrient-restricted tumor microenvironment where glucose is relatively low. As glucose levels in PDAC interstitial fluid are reported to be in the low millimolar range (PMID: 30990168), it would be informative to repeat the HBP metabolite measurements at higher glucose levels than those used in the present version of the manuscript and assess whether PDAC cells can maintain UDP-GlcNAc levels under these less stringent but physiologically relevant conditions (perhaps together with glutamine restriction to induce NAGK), and to perform proliferation assays in control versus NAGK knockdown/knockout cells under these glucose-low conditions.

6. Reviewers requested additional details for the Materials and methods section. Cell seeding numbers and incubation times prior to sample collection should be added to metabolite quantitation and labeling, Western blotting, RT-qPCR and lectin binding assay sections. The normalization methods used for metabolite quantification and isotope correction should be provided.

---

## [Author Response]

Essential revisions:1. The reviewers expressed some confusion with isotope tracing data presented in Figure 2 and Figure S2. Figure S2 shows no apparent M+6 in GlcNAc-P (in contrast to what is shown in Figure 2) and a similar amount of M+6 enrichment in UDP-GlcNAc in both high and low glutamine conditions, contrasting with reduced M+6 following glutamine deprivation in Figure 2C. Is it possible that Figure 2C might be showing M+8 for GlcNAc-P (middle) and M+5 for UTP (bottom), instead of what is listed? A related question based on Figure 2C is why, under glutamine restriction, GlcNAc-P labeling is completely abolished but labeling of UDP-GlcNAc (which is downstream) is only decreased by about half?Given the potential issues with these data, it is hard to evaluate the conclusion that glucose tracing to UDP-GlcNAc is reduced during conditions of low glutamine. Importantly, Figure S2C appears to show that both M+6 and M+8 of UDP-GlcNAc is maintained during low glutamine, raising the possibility that synthesis of the glucosamine ring is not disrupted by low glutamine, which appears based on Figure S2 to specifically reduce synthesis of UTP. Given the critical importance of these tracing data to the central model of the paper-namely, that glutamine restriction reduces de novo GlcNAc synthesis-it is important for the authors to address these concerns. Additional concerns raised by the isotope tracing data are listed below.

We thank the reviewers for these comments- these are critical points and we appreciate the opportunity to clarify and expand on these central data.

1) Data representation: The data shown in Figure 2 do not represent the M+6 isotopologue solely, but rather the sum of UDP-GlcNAc isotopologues containing the labeled glucosamine ring (i.e., adding M+6, M+8, M+11, and M+13). To make this clear, we have a) changed the labeling on the graphs in Figure 2D to reference the part of the molecule that is labeled, b) provided text edit to explicitly describe how the data is represented, and c) added additional labeling of the graphs in Figure S2.1C-D. We hope the reviewers find the new presentation of the data to be clear.

2) Discrepancy between GlcNAc-P and UDP-GlcNAc labeling: Thank you for pointing this out. We have now repeated these experiments in PANC-1 cells, as well as in two other PDA cell lines, and we find that while there are some slight differences in the timing of labeling into GlcNAc-P and UDP-GlcNAc, the labeling tracks much more closely than that reported in the initial submission (revised Figure 2D and S2.1C-D). We believe the initial result of complete loss of GlcNAc-P labeling in low glutamine reflected technical limitations in detection. We have further optimized metabolite extraction efficiency, and signal has improved. Now, in multiple repeats in several cell lines, we see markedly suppressed but detectable labeling of GlcNAc-P under glutamine restriction, and this tracks closely with UDP-GlcNAc labeling. Nevertheless, there are subtle differences in the timing of GlcNAc-P and UDP-GlcNAc labeling, particularly in PANC-1 cells where GlcNAc-P labeling under glutamine restriction is very low. To consider the possibility of a minor GlcNAc-P-independent route to UDP-GlcNAc synthesis from glucose, we have also conducted experiments involving silencing of the 2 major pathways that branch off from UDP-GlcNAc to generate UDP-GalNAc or sialic acid. In particular, we hypothesized that the reversible enzyme GALE (UDP-galactase-4-epimerase), which interconverts UDP-GlcNAc and UDP-GalNAc, would be a plausible candidate to enable entry of glucose carbon into the UDP-GlcNAc pool in a manner independent of GlcNAc-P, although such a route has not been annotated to our knowledge. We also tested GNE (bifunctional UDP-N-acetylglucosamine 2-epimerase/N-acetylmannosamine kinase), which mediates the first step towards CMP-sialic acid synthesis from UDP-GlcNAc. Glucose-dependent labeling of UDP-GlcNAc was not reduced, however, by silencing of either enzyme in high or low glutamine conditions (revised Figure S2.2D). Thus, the slight difference in incorporation is most likely due to a difference in detection, since UDP-GlcNAc is much more abundant than GlcNAc-P, rather than a bypass route, though we cannot rule out the possibility of a minor contribution from an unknown pathway. We have revised the text to state this. Regardless, the core conclusion that de novo hexosamine synthesis from glucose is suppressed in low glutamine conditions is strongly supported by the data.

3) Labeling of ribose component of UDP-GlcNAc: In response to the reviewer query, we have also further explored the interesting observation that in addition to reduced de novo synthesis of the glucosamine component of UDP-GlcNAc, labeling of the ribose component is also strikingly reduced in low glutamine conditions (revised Figure 2D, S2.1C, S2.2A). We now explicitly highlight this in Figure 2D. These data suggest that nucleoside salvage is also triggered under low glutamine conditions, which is consistent with findings of a prior study that demonstrated that autophagy sustains uridine phosphate pools in the absence of glutamine (Guo et al., Genes Dev, 2016). Uridine is salvaged via the enzymes uridine-cytidine kinase 1 and 2 (UCK1/2), both of which are expressed in PDA cells. We have used RNAi to silence UCK1 or UCK2, finding that silencing of either enzyme reduces uridine phosphate and UDP-GlcNAc pools (revised Figure S2.2B-C). Thus, the data now clearly indicate that de novo hexosamine and nucleotide synthesis are suppressed in the context of glutamine restriction.

2. In Figure 1, the authors show that protein glycosylation is preserved in PDAC cells even under low glutamine conditions. Given that PDAC cells are reported to be quite reliant on glutamine availability, is the viability of these cells affected under these growth conditions? Can the authors design experiments to determine whether the modification is maintained because of sustained deposition of GlcNAc or because of changes in cell survival/protein turnover and/or removal of glycoslyation?

This is an excellent point, and we have conducted new experiments to address this point. In revised Figures 1G and S1F, we have explicitly tested the ability of cells to continue to deposit GlcNAc by examining changes in O-GlcNAcylation in the presence of the O-GlcNAcase inhibitor Thiamet G. We demonstrate that under low glutamine conditions, O-GlcNAcylation still actively occurs in low glutamine and to a similar extent to that observed in high glutamine conditions. This supports the model that the modification is maintained at least in part because of sustained deposition of GlcNAc. We have also blotted for cleaved PARP-1 in these experiments to assess viability (Figures 1G and S1F). Additionally, we point out that, as stated in the methods, for all flow experiments we used DAPI or π staining to gate out dead cells, so the results in Figure 1 for LPHA staining are representative of live cells.

3. It will be important for the authors to investigate whether NAGK deletion affects levels of glycosylation in cultured cells (especially during culture under low glutamine) and, if possible, in tumor samples. Figure 3D shows that UDP-GlcNAc pools are equivalent in NAGK deleted cells cultured in low or high glutamine, which at face value argues against NAGK as an important source of UDP-GlcNAc during glutamine restriction, but this interpretation could be affected by changes in overall cellular capacity for protein glycosylation.

We have conducted several experiments to examine the effects of NAGK deletion on glycosylation. In tumor lysates, the LPHA signal is elevated in NAGK KO as compared to control tumors (Figure S4D). While this is converse to what might have been anticipated, it is consistent with the model that NAGK impacts glycosylation. The increased signal may reflect upregulation of de novo synthesis in the cancer cells within the small tumors that form in the absence of NAGK, compensatory regulation of glycosylation turnover, or high LPHA signal from other cells within the tumor. We have also conducted experiments examining glycosylation in cultured cells as well. Unfortunately, results were inconsistent from clone to clone, with some but not all NAGK KO clones showing reduced L-PHA binding. While it is disappointing that results were not as clear cut as we would have hoped, the regulation of glycosylation is complex and will require substantial additional work beyond this manuscript.

4. As NAGK loss-of-function experiments are limited to the PANC-1 cell line, the authors should repeat a few key experiments in an additional NAGK knockdown/knockout PDAC cell line.

We have generated NAGK knockout MIA PaCa-2 cell lines, and repeated key experiments, which are now included in revised Figures 3 and 4 and corresponding supplemental figures.

5. The authors nicely demonstrate that glutamine deprivation reduces flux through the HBP while inducing NAGK and enabling UDP-GlcNAc maintenance via GlcNAc salvage. The PDAC microenvironment also features restricted levels of glucose, such that PDAC cells face reduced levels of both nutrients relevant to the HBP in intact tumors. However, the impact of glucose restriction on UDP-GlcNAc pools presented in Figure S2B is somewhat difficult to reconcile with the compelling results of in vivo studies presented in Figure 4E and F, which suggest a critical role for NAGK to support PDAC cell proliferation within a nutrient-restricted tumor microenvironment where glucose is relatively low. As glucose levels in PDAC interstitial fluid are reported to be in the low millimolar range (PMID: 30990168), it would be informative to repeat the HBP metabolite measurements at higher glucose levels than those used in the present version of the manuscript and assess whether PDAC cells can maintain UDP-GlcNAc levels under these less stringent but physiologically relevant conditions (perhaps together with glutamine restriction to induce NAGK), and to perform proliferation assays in control versus NAGK knockdown/knockout cells under these glucose-low conditions.

In revised Figure S2.1B, we quantified HBP metabolites in 5 mM glucose conditions and observed that UDP-GlcNAc abundance was reduced compared to that in 10 mM glucose. While further investigation is certainly required, the quantification of UDP-GlcNAc in 5 mM and 0.1 mM glucose suggests that UDP-GlcNAc levels are more responsive to changes in glucose than changes in glutamine, and that the salvage pathway may be most important under conditions of glutamine restriction. We hypothesize that in addition to NAGK regulation, this is due to increased free GlcNAc available for salvage under glutamine restriction, which we now report in revised Figure 3A. Further investigation will be needed to ascertain the sources of GlcNAc under glutamine restriction (lysosomal breakdown of glycoconjugates is one possibility, but will require additional study). In the manuscript co-submitted with this one by Lyssiotis and colleagues (https://www.biorxiv.org/content/10.1101/2020.09.14.293803v1), hyaluronic acid is proposed as a key source of GlcNAc in the tumor microenvironment. Taking the two studies together, it is likely that salvage is facilitated under conditions in which free GlcNAc abundance is elevated. We have added discussion to this point to the text.

6. Reviewers requested additional details for the Materials and methods section. Cell seeding numbers and incubation times prior to sample collection should be added to metabolite quantitation and labeling, Western blotting, RT-qPCR and lectin binding assay sections. The normalization methods used for metabolite quantification and isotope correction should be provided.

Clarification on these aspects of the methods has been added to the methods sections.